# Heart Rate Variability at Rest Predicts Heart Response to Simulated Diving

**DOI:** 10.3390/biology12010125

**Published:** 2023-01-13

**Authors:** Krzysztof S. Malinowski, Tomasz H. Wierzba, J. Patrick Neary, Paweł J. Winklewski, Magdalena Wszędybył-Winklewska

**Affiliations:** 1Department of Physiology, Faculty of Medicine, Medical University of Gdansk, 80-210 Gdansk, Poland; 2Faculty of Kinesiology & Health Studies, University of Regina, Regina, SK S4S 0A2, Canada; 3Department of Human Physiology, Faculty of Health Sciences, Medical University of Gdansk, 80-210 Gdansk, Poland; 4Institute of Health Sciences, Pomeranian University of Slupsk, 76-200 Slupsk, Poland

**Keywords:** diving, simulated diving test, heart rate, heart rate variability, healthy individuals

## Abstract

**Simple Summary:**

The diving reflex is a complex response of the cardiovascular system that allows mammals, including humans, to survive immersion in water, as well as hypoxia caused by respiratory arrest. The response is adaptive, preferentially protecting brain tissue from the effects of apnea-induced hypoxia. In everyday life, there are situations in which there is a temporary apnea with simultaneous cooling of the face. This can trigger a hemodynamic response with an increase in blood pressure and a slow heart rate as a result of the diving reflex mechanism. Diving response is mediated by the autonomic nervous system with simultaneous extensive stimulation of the sympathetic and parasympathetic systems, which can evoke life-threatening arrhythmias. A characteristic feature of the cardiac response to diving is the uncertainty in predicting an individual’s outcome. The current research examined the poorly understood regulatory oscillations of the heart rhythm and their influence on the course of the cardiac response to diving. The results of the research indicate that the cardiac response to diving is strictly dependent on the autonomic regulation of the heart rhythm under resting conditions. The present work provides a foundation for further research to preventative measures that could cause unfavorable course of cardiodepressive responses.

**Abstract:**

A characteristic feature of the cardiac response to diving is the uncertainty in predicting individual course. The aim of the study was to determine whether resting regulatory heart rate determinants assessed before diving may be predictors of cardiac response in a simulated diving test. The research was conducted with 65 healthy volunteers (37 women and 28 men) with an average age of 21.13 years (20–27 years) and a BMI of 21.49 kg/m^2^ (16.60–28.98). The simulated diving test consisted of stopping breathing after maximum inhaling and voluntarily immersing the face in water (8–10 °C) for as long as possible. The measurements included heart rate variability (HRV) analysis before diving and determination of the course of the cardiac response to diving—minimum and maximum heart rate (HR). The results indicate that minimum HR during diving (MIN_div) is dependent on the short-term HRV measures, which proves the strong influence of the parasympathetic system on the MIN_div. The lack of dependence of MIN_div on short-term HRV in women may be associated with differences in neurogenic HR regulation in women and men. In conclusion, cardiac response to simulated diving is strictly dependent on the autonomic regulation of the heart rhythm under resting conditions. The course of the cardiac response to diving and its relationship with resting HRV appears to be gender dependent.

## 1. Introduction

Diving is a popular activity that is undertaken for recreational and sport purposes. Freediving is a form of underwater diving that relies on breath-holding until resurfacing rather than using a breathing apparatus, such as scuba gear. Breath-hold diving triggers a complex reflex response known as the diving response, the aim of which is to increase the possibility of human survival in the aquatic environment [1]. During diving, an organism has to cope with a drop in the partial pressure of oxygen (PO_2_) in arterial blood. The risk of insufficient oxygen supply is the appearance of disturbances in cell functioning and even cell death. The retrograde changes caused by oxygen deficiency are associated with the reduction or complete inhibition in oxidative phosphorylation. Cells characterized by a living metabolism, including nerve cells and cardiomyocytes, are exposed the most to hypoxia [2]. During diving, autonomous mechanisms are triggered. The diving response is redistribution of blood with oxygen for vital organs and is therefore referred to as an oxygen-conserving mechanism [3].

The diving response is a complex reflex initiated by apnea and cooling of the facial skin, especially the vestibule of the nose and the forehead. The response is triggered by stimulation of arterial chemoreceptors and trigeminal nerve endings located in the skin [4]. This triggers the arterial chemoreceptor reflex and the synergistic trigeminocardiac reflex [5]. The body’s reaction to diving is mediated by the autonomic nervous system (ANS) with simultaneous stimulation of the sympathetic and parasympathetic systems. The stimulation of the intracardiac parasympathetic nerves slows down the heart rate (HR) and lengthens the conduction time of the depolarization wave from the atria to the ventricles. At the same time, the intracardiac sympathetic innervation is activated, which has the opposite, positive chronotropic effect [6]. However, there is prevalence of the vagus nerve on heart rhythm, which can be associated with bradycardia with HR near 20–30 beats per minute [7]. On the other hand, sympathetic stimulation is responsible for constriction of peripheral blood vessels and the associated increase in total vascular resistance, which leads to an increase in blood pressure with simultaneous redirection of blood to the body’s core. Oxygen supplies are directed to the hypoxia-sensitive brain and heart at the expense of ischemia and hypoxia of the abdominal organs and large oxygen-consuming skeletal muscles. Experienced divers, e.g., pearl divers, mainly Japanese women called AMA divers, can spend up to five minutes underwater thanks to the diving response [8].

Under laboratory conditions, the diving response may be tested by a simulated diving test. It consists of a maximum inhale and submerging the face in the water for the time specified by the research protocol. A characteristic feature of the cardiac response to diving is HR reduction shortly after submerging. However, an increase in HR shortly before the test can be observed, which is a symptom of emotional agitation associated with intracardiac sympathetic activity [9,10].

HRV analysis was used as a tool for assessing the influence of the autonomic nervous system on the heart rhythm, and thus determining the role of neurogenic regulation on the cardiac response to diving. HRV is an established and non-invasive research method that describes the effects of the autonomic nervous system on HR. Thus, it is a tool for determining the profile of neurogenic, extrinsic heart regulation and its influence on HR while diving. Resting heart rhythm is influenced by the sympathetic and parasympathetic activity of the ANS, which determines the short- and long-term variability of sinus rhythm. Moreover, total HRV is the result of other endogenous biological oscillators that include respiratory movements, fluctuations in blood pressure, and hormonal influences [11,12].

The parameters of total HRV reflect the complexity of the regulatory mechanisms of the cardiovascular system. It is widely believed that a reduction in HRV is associated with a reduction in the complexity of control mechanisms, which leads to “stiffening” of the HR, associated with greater regularity of heart rhythm and reduced responsiveness to emerging disturbances in homeostasis [13,14].

Disease processes, aging, lifestyle, the influence of the external environment or neuropsychological conditions can lead to disturbances in the autonomic regulation of heart rhythm and thus affect the variability of HR [13]. There are many reports in the literature of a close relationship between decreased HRV parameters and cardiovascular diseases. HRV reduction is associated with an increased risk of death in post-myocardial infarction patients. In addition, there is a relationship between HRV parameters and classic risk factors for coronary artery disease, such as arterial hypertension and atherosclerosis [15,16,17].

A highly individualized and unpredictable course of the reflex response prompted the authors of the study to explore the relationships between the course of the cardiac response to diving with parameters of the HRV at rest in the time and frequency domains [4,18]. Both at rest and during the simulated diving test, HR is under the neurogenic control of the ANS. It can be assumed than that the indicators of the autonomic regulation of heart rhythm–HRV indices may correlate with the cardiac response during diving [4]. The primary objective of this study was to assess the maximum and minimum HR during the simulated diving test and study the relationship between them and the HRV indexes calculated from the resting electrocardiogram (ECG) recording. The maximum HR during diving corresponds to the anticipatory HR acceleration associated with sympathetic innervation of the heart. Minimum HR during diving is the equivalent of the parasympathetic influences on the heart that are responsible for the cardiodepressive response of the simulated diving test. It was therefore assumed that the maximum and minimum HR during diving could be used as an indicator of the functional state of the autonomic nervous system during the test [19].

We hypothesized an especially strong relationship between short-term HRV and minimum HR during the diving test, and therefore, resting short-term HRV indices could serve as a predictor of cardiodepressive response during diving. Additionally, we assumed that the relationship of HRV indices with cardiac response to diving is gender-specific (different in males and females).

Finding an association between the HR during the simulated diving test and HRV determinants at rest may be useful in the employment of HRV in predicting cardiac response to diving.

## 2. Materials and Methods

The conducted research was of a cognitive nature and did not pose a threat to the health of the subjects. The study was carried out with healthy volunteers, recruited among students of the Medical University of Gdańsk. After completing the health questionnaire and medical history, the subjects were instructed about the details of the study. Bodyweight, height, and blood pressure were measured. The exclusion criteria were based on the data from the questionnaire and medical history. Diseases of the respiratory and cardiovascular systems (e.g., cardiac arrhythmia, resting tachycardia, syncope, and hypertension) were exclusion criteria. Disclosure of abnormalities in the ECG during the initial resting recording resulted in exclusion from participation in the study.

The study was conducted in accordance with the Declaration of Helsinki. The protocol was approved by the Independent Bioethics Commission for Research at the Medical University of Gdansk (NKBBN/471/2013). All participants were informed about the procedures, risks, and expected outcomes before starting the experimental procedure and gave their written informed consent for participation.

This study was conducted with 65 healthy volunteers, including 37 women and 28 men. The participants’ mean age was 21 years, and ranged from 20 to 27 years. Body mas index (BMI) was 21.49 kg/m^2^ (16.60–28.98).

The main element of the experimental procedure was immersion of the face in low-temperature water, combined with a voluntary, longest apnea-simulated diving test. Before starting the test, the subject remained seated for 10 min with elbows and forearms resting on the table. At a given sign, the subject performed a maximal breath and then submerged his/her face in the water (8–10 °C), trying to stay submerged for as long as possible. Before and during the simulated diving test, each participant’s ECG was recorded at a sampling rate of 4 kHz using an integrated data acquisition system (ADInstruments Research System, AdInstruments, New Zealand). The examinations were carried out in the presence of an experienced specialist in internal medicine (THW), who continuously monitored the regularity of the heart rate, immediately stopping the test in the event of cardiac arrhythmias during the prolonged diving. All subjects completed the simulated diving test.

### 2.1. Heart Rate Analysis

The analysis of the cardiac response to diving consisted in determining of the time series of the identified R waves. On this basis, the durations of the RR intervals (RRi) were determined. Then the longest and shortest RRi were used to calculate the minimum (MIN_div) and maximum (MAX_div) HR during diving.

### 2.2. HRV Analysis

The heart rate variability analysis was calculated from a 512 RRi segment of the resting ECG using statistical methods and the fast Fourier transform (FFT) method using Kubios HRV Pro software (Kuopio, Finland). Assessment of total heart rate variability was based on the standard deviation of NN (normal to normal) intervals (SDNN) and on the total spectrum power (TP ms^2^) of the HRV frequency analysis. Short-term HRV indices, such as root mean square of successive differences (rMSSD) and pNN50% (the proportion of NN50 divided by total number of NNs), were used to reflect parasympathetic cardiac activity [12]. The power of the spectrum in the high frequency range (HF ms^2^) corresponds to changes in HR with a frequency of 9–24 times per minute (0.15–0.4 Hz). HF (ms^2^) was used as an indicator of the activity of the parasympathetic intracardiac nerves in the HRV frequency analysis. The power of the spectrum in the low frequency range (LF ms^2^) reflects changes in HR with a frequency of 2.4–9 times per minute (0.04–0.15 Hz). It was used to assess the long-term heart rate variability.

### 2.3. Statistical Analysis

The nature of the data distribution was assessed with the Shapiro–Wilk test. Depending on the distribution of the analysed data, parametric or non-parametric tests were used, and the results were presented as mean ± SD. The mean values of the independent samples (man/woman) were compared using the Student’s *t*-test or U Mann–Whitney test. The relationships between pairs of independent variables were analysed using the Pearson and Spearman linear regression method. A significance level of *p* < 0.05 was considered significant. The significance level of statistical analysis was calculated for two-tailed. The HRV analysis data were divided into two subgroups: less and equal (_≤MED_) and greater (_>MED_) than the median. The median was used to obtain numerically similar subgroups of data, while the division into subgroups was used for a more detailed analysis of the relationship between the minimum and maximum HR during diving and the selected parameters of the HRV analysis in the time and frequency domain. All statistical calculations were performed using statistical package Statistica 10 (StatSoft Inc., Tulsa, OK, USA).

## 3. Results

### 3.1. Anthropometric Data

Table 1 presents anthropometric data, including age, body mass, height, body mass index (BMI), and body water and body fat percentages.

### 3.2. HR Response to Simulated Diving

Immersion in water resulted in HR acceleration followed by a cardiodepressive response (Table 2). The maximum HR during diving was 112.98 ± 16.23 min^−1^ and was higher in women (117.44 ± 16.38 min^−1^ vs. 107.09 ± 14.25 min^−1^). In all subjects, we noticed a decrease in HR during the simulated diving test. The minimum HR during diving (MIN_div) was 52.66 ± 8.16 min^−1^ and was significantly lower in men compared to women (49.35 ± 7.73 min^−1^ vs. 55.17 ± 7.64 min^−1^). In the current study, arrhythmias occurred rarely and concerned only a few cases (five times). It concerned single ventricular extrasystoles (three times) and partial lengthening of atrioventricular conduction (two times). In each case of arrhythmia, the simulated diving test was stopped immediately and the heart rhythm returned to regular. Since arrhythmias appeared at the end of the diving, there were no reasons to exclude these trials from the study.

### 3.3. HRV Analysis

HRV measures in the time domain in the resting state are presented in Table 3 and Table 4. Total heart rate variability, as measured by SDNN, was 60.51 ± 22.37 ms. There was a significant difference in the subgroup of values higher than the median -SDNN_>MED_. In the group of men, it was 87.28 ± 21.09 ms, and in the women, it was 69.54 ± 13.48 ms.

The short-term variability indexes were 44.78 ± 24.02 and 19.92 ± 16.51 for rMSSD and pNN50%, respectively. There were no significant differences between women and men independent of the division into subgroups.

Table 5 and Table 6 represent HRV measures in the frequency domain in the resting state. Measures of total heart rate variability in the frequency domain analysis are the total range of power spectral density, total power (TP). TP (ms^2^) was 4016.13 ± 3099.80 ms^2^ and was higher in the men subgroup of values higher than the median. TP_>MED_ was 7744.70 ± 3683.16 ms^2^ in men and 4905.17 ± 2027.73 ms^2^ in women.

The low frequency power component, LF (ms^2^), was 2269.90 ± 1857.76 ms^2^. There was also a significantly higher value of the LF measure in the men subgroup of values higher than the median (LF_>MED_). It was 4389.88 ± 2108.79 ms^2^ in men and 2830.73 ± 1521.06 ms^2^ in women.

The high frequency power component, HF (ms^2^), was 1177.15 ± 1288.87 ms^2^. There were no significant differences between men and women.

The SD_1_ and SD_2_ measures were 31.79 ± 17.02 and 99.60 ± 34.39, respectively. SD_2_ differed between men and women in the subgroup of values higher than the median. SD_2>MED_ was 140.90 ± 30.55 in men and 114.73 ± 15.26 in women.

### 3.4. Dependence of Cardiac Response to Diving on Resting HRV

The correlation coefficients between cardiac response to diving (MIN_div; MAX_div) and the selected resting HRV measures in the time-domain analysis are presented in Table 7. The results indicate that MIN_div is dependent on the short-term HRV measures. This relationship is inversely proportional; an increase in short-term variability is accompanied by a decrease in MIN_div. The correlations of MIN_div with rMSSD and pNN50% were highest in the men subgroup whose rMSSD and pNN50% values were higher than the median, rMSSD_>MED._; and pNN50%_> MED._ The correlation coefficients for these pairs were −0.7758 and −0.6556, respectively. The dependence of MIN_div on short-term HRV is confirmed by the results of the correlation with HRV measures in the frequency domain analysis and with the use of non-linear methods of assessing heart rate variability (Table 8).

There were high correlation coefficients between MIN_div and short-term HRV represented by HF (ms^2^) and SD_1_. In the men subgroup whose HF (ms^2^) and SD_1_ values were higher than the median (HF_>MED._; SD_1>MED._), the correlation coefficients were −0.6263 and −0.7758, respectively. The relationships of the maximum HR during the dive (MAX_div) with the measures of total, short- and long-term heart rate variability did not show significant correlation coefficients.

## 4. Discussion

There are two new findings of this study: (1) the short-term variability of the HR before the test is a predictor of a cardiodepressive reaction during diving and (2) the heart response to simulated diving differs between men and women.

Christoforid et al. assessed the activity of the intracardiac nerves of the ANS in professional divers [20]. Thirteen swimmers with several years of free diving experience (without oxygen apparatus) and thirteen people without swimming experience (control group) participated in the study. The HRV analysis was performed using a 24 h ECG record. In the group of divers, significantly higher values of all HRV indices were obtained. The total variability of the heart rate expressed by SDNN and TP (ms^2^) was higher in the group of divers by 38.1% and 7.2%, respectively. On the other hand, rMSSD and HF (ms^2^), describing short-term heart rate variability, were higher by 61.23% and 74.9%, respectively. The results of the study also show a 23.9% lower minimum heart rate and a 20.6% lower average heart rate at rest in athletes. The slowest HR was recorded in one of the study participants during the night hours, 32 min^−1^. The above-mentioned study may be a premise for the relationship between high activity of intracardiac ANS at rest and better adaptation of the body to diving. Moreover, short-term variability of HR at rest can be an expression of the regulatory potential of parasympathetic heart control.

Our study’s results indicate the dependence of the slowest heart rate during the dive, MIN_div, on short-term variability at rest, expressed both by HRV indices in the time and frequency domains. There was a high, negative correlation coefficient between MIN_div and the rMSSD, pNN50%, HF, or SD_1_ index. This means there is a close relationship between intracardiac activity of the parasympathetic system at rest and the minimum HR during the diving test. It is especially noticeable in subjects with high values (greater than the median) of short-term variability—rMSSD_>MED._, pNN50%_>MED._, HF_>MED._, or SD_1>MED_. The highest correlation coefficient was noted in men with high rMSSD values (rMSSD_>MED._). The correlation coefficient for the relationship of MIN_div–rMSSD_>MED._ in men was −0.7758. Lower MIN_div values were noted in men compared to women (49.35 ± 7.73 min^−1^ vs. 55.17 ± 7.64 min^−1^). Thus, it can be concluded that the greater activity of the intracardiac fibres of the vagus nerve at rest, the greater reduction in HR during the simulated diving test, and the short-term variability of the HR before the test may be predictors of a cardiodepressive reaction while diving

The lack of dependence of MIN_div on the determinants of short-term variability of HRV (before the diving test) in women may be associated with differences in neurogenic heart rate regulation in women and men. It should be noted that there was significantly higher minimum and maximum HR during the diving test in women, which can be an expression of lower activity of the parasympathetic system in the cardiodepressive reaction and a higher influence of the sympathetic system on the heart in the anticipatory effect.

The HRV analysis indicated differences in the spectral power density for short- and long-term variability between men and women. Koenig and Thayer conducted a meta-analysis comparing the HRV indices in men and women in 172 studies [21]. The results of the studies indicate a higher mean HR in women along with lower mean values of RR intervals. Women showed a lower total variability of heart rhythm, represented both by index of the time domain, SDNN, as well as the frequency domain, TP (ms^2^). Further differences relate to the power of the spectral density of the low and high oscillations of HRV. Lower LF rates and higher HF were reported in women compared to men. These results indicate a higher activity of intracardiac parasympathetic activity in women, which in turn is in contradiction to the statistically higher HR at rest in this group of respondents. However, the cited meta-analysis included studies that differed in ECG recording time, as well as the position of the examined person and the frequency of breathing. Research procedures and the method of HRV analysis are of great importance for the determined indicators and the interpretation made on its basis [22].

In the present study, higher short-term variability (rMSSD, pNN50%, HF, or SD_1_) and higher total variation (SDNN) in men was observed. However, the lack of statistical significance of differences between women and men does not allow us to attribute higher intracardiac parasympathetic activity in men. The literature does not provide unequivocal answers regarding the differences in HRV indicators in women and men, which indicates the need for further research in this area.

The increase in HR observed before the test is called the anticipatory effect. The increase in HR may persist for a few seconds after the face is immersed in water, and it is associated with the body’s response to the stress associated with the performed test [8]. The stimulation of the intracardiac sympathetic nerves accelerated the heart rhythm to 112.98 ± 16.23 min^−1^. The obtained results indicate a significantly higher anticipatory effect in women (117.44 ± 16.38 min^−1^ vs. 107.09 ± 14.29 min^−1^). There was no direct correlation between parasympathetic activity (expressed by HRV indices of short-term variability) and MAX_div. This is probably due to the dependence of an anticipatory effect on sympathetic activity. Moreover, no correlation was found between MAX_div and any HRV indices, which does not exclude the participation of increased sympathetic activity on MAX_div, but may result from no clear reflection of sympathetic activity by HRV analysis indicators [23].

According to most of the literature data, the standard cardiac response to a simulated diving test is a reduction in HR [24]. Interesting data are presented in the meta-analysis by Schipke et al., where the cardiodepressive reaction in response to diving was analysed. Based on the inclusion and exclusion criteria, eight studies were selected with a total of 182 subjects aged 19–27 who performed a simulated diving test in similar conditions and according to a similar procedure to that performed in this study. The results indicate that the calculated average heart rate reduction during the test was 59 beats per minute (bpm) and ranged from 48 to 67 bpm [25]. There is also a report on the extreme slowing of HR, up to five beats per minute, recorded in a 41-year-old man during apnea combined with immersion of the face in water at a temperature of 2 °C [26]. In this study, MIN_div was 52.66 ± 8.16 min^−1^. The low temperature of the water in which the subjects immersed their faces potentially had a significant impact on the cardiac response, which was associated with the enhancement of the cardiodepressive response evoked by the trigemocardial reflex [27].

The LF band was originally thought to be an approximate indicator of cardiovascular sympathetic activity. The research related to the spectral analysis during the orthostatic tests shows the reduction in the HF band with a simultaneous increase in the LF component in the power spectrum. It is associated with an increase in the activity of the sympathetic nervous system in response to the decompression of baroreceptors [28]. Additionally, an increase in the amplitude of the LF band is observed during sympathetic arousal under conditions of mental stress or exercise [29]. However, the assumption of a simple relationship between the activity of intracardiac sympathetic activity and the variability of the heart rate in the low frequency range was not confirmed in further studies. It was shown that activity of intracardiac parasympathetic fibres may also contribute to the LF component. This is evidenced by studies related to the blockade of muscarinic receptors, M_1_, which leads not only to the reduction in the HF band, but also to the reduction in the low-frequency component of HRV [30]. It is also suggested that the LF oscillations are an expression of the activity of the efferents of the sympathetic and parasympathetic systems, modulated by changes in arterial pressure in the arterial baroreceptor reflex [31]. The influence of the ANS on heart rate variability in the low frequency range is unquestionable, while the interpretation of the “physiological correlates” of the LF band encounter difficulties due to the complexity of the regulatory mechanism of heart rhythm.

In our study, the investigated population consisted of healthy medical students. We did not measure VO_2_, so we are not able to provide their detailed training capacity, but volunteers did not declare practising any particular sport, and the students most likely represented the typical population of students of average physical activity. In a few cases of arrhythmia (five times), the simulated diving test was stopped. Since arrhythmias appeared at the end of the diving, there were no reasons to exclude these participants from the study.

## 5. Conclusions

In everyday life, there are situations in which there is a temporary apnea with simultaneous cooling of the face. This can trigger a hemodynamic response with an increase in blood pressure and a slow HR in the mechanism of the diving reflex. Sudden cooling of the face, especially in the nasal vestibule, associated with temporary inhibition of expiration or apnea, may trigger a reflex cardiodepressive response.

The results of present study indicate that selected indicators of the neurogenic regulation of HR at rest can be used to approximate the course of the cardiac response during diving and could open the field for practical applications. The development of warning algorithms against the unfavourable course of the cardiodepressive response to diving could be used to prevent their unfavourable course. Prospectively, new diagnostic algorithms could be incorporated as part of the software for sensors such as a heart rate monitor, which are increasingly used during recreational and sports physical activity, as well as when performing professional work in conditions similar to those that occur while diving.

The previously cited studies suggest a different regulatory profile of the heart rhythm in athletes. Normally, increased parasympathetic activity could predispose them to greater HR reduction during diving. Additionally, apnea time, depending on training and on physical predispositions (especially on total lung capacity), can also affect a different cardiac response to diving. The above premises may be the subject of further studies examining the impact of training and fitness on the cardiac response to diving and its relationship with resting HRV. Moreover, interesting studies could refer to an explanation of the detailed impact of the trigeminal and arterial chemoreceptors reflex on the cardiodepressive response to diving.

The cardiac response in the simulated dive test is dependent on the intracardiac influences of the ANS. The varying strength of the parasympathetic and sympathetic nervous systems leads to different cardiac responses during diving, including potential arrhythmias. Further research could also focus on the relationship between HRV, illustrating the influence of ANS on the heart rhythm, and spontaneously occurring arrhythmias.

## Figures and Tables

**Table 1 biology-12-00125-t001:** Anthropometric data.

	All	Women	Men	P [w vs. m]
Age	21.13 ± 1.34	21.03 ± 0.91	21.30 ± 1.89	0.4700 _NS_
Body mass (kg)	65.12 ± 11.25	59.25 ± 7.18	74.73 ± 10.08	0.0000 ***
Height (m)	1.74 ± 0.09	1.69 ± 0.05	1.82 ± 0.08	0.0000 ***
BMI—body mass index (kg/m^2^)	21.49 ± 2.33	20.81 ± 2.03	22.62 ± 2.39	0.0033 ***
%Fat (%)	22.05 ± 7.80	23.70 ± 5.38	18.76 ± 10.64	0.0316 *
%H_2_O (%)	54.28 ± 3.93	52.67 ± 3.36	57.70 ± 2.68	0.0000 ***

* *p* < 0.05; *** *p* < 0.001; and NS—not significant.

**Table 2 biology-12-00125-t002:** The HR response to simulated diving in healthy men and women.

	All	Women	Men	P [w vs. m]
MIN_div (min^−1^)	52.66 ± 8.16	55.17 ± 7.64	49.35 ± 7.73	0.0036 **
MAX_div (min^−1^)	112.98 ± 16.23	117.44 ± 16.38	107.09 ± 14.25	0.0098 **

** *p* < 0.01.

**Table 3 biology-12-00125-t003:** Analysis of heart rate variability—HRV in the time domain.

	All	Women	Men	P [w vs. m]
SD_NN (ms)	60.51 ± 22.37	56.48 ± 17.07	65.83 ± 27.31	0.0953 _NS_
rMSSD (ms)	44.78 ± 24.02	41.33 ± 18.63	49.34 ± 29.44	0.5762 _NS_
pNN50% (%)	19.92 ± 16.51	19.38 ± 15.23	20.63 ± 18.33	0.9735 _NS_

NS—not significant.

**Table 4 biology-12-00125-t004:** Analysis of heart rate variability—HRV in the time domain for groups divided by the median.

		All	Women	Men	P [w vs. m]
SD_NN (ms)	_≤MED._	43.80 ± 8.88	44.11 ± 8.90	44.38 ± 10.64	0.9854 _NS_
	_>MED._	77.74 ± 18.65	69.54 ± 13.48	87.28 ± 21.09	0.0115 *
rMSSD (ms)	_≤MED._	27.87 ± 7.65	27.90 ± 7.79	27.93 ± 7.92	0.9274 _NS_
	_>MED._	62.22 ± 22.63	55.51 ± 16.01	70.74 ± 27.41	0.1744 _NS_
pNN50% (%)	_≤MED._	7.90 ± 5.51	8.07 ± 5.80	6.46 ± 4.42	0.6488 _NS_
	_>MED._	33.94 ± 13.72	31.32 ± 12.72	34.80 ± 15.68	0.8345 _NS_

* *p* < 0.05; and NS—not significant.

**Table 5 biology-12-00125-t005:** Analysis of heart rate variability—HRV in the frequency domain.

	All	Women	Men	P [w vs. m]
Total Power (ms^2^)	4016.13 ± 3109.80	3368.07 ± 2112.47	4872.51 ± 3954.41	0.3236 _NS_
LF (ms^2^)	2269.90 ± 1857.76	1867.00 ± 1436.92	2802.30 ± 2216.16	0.0645 _NS_
HF (ms^2^)	1177.15 ± 1288.87	1000.80 ± 886.21	1410.18 ± 1670.79	0.8998 _NS_
SD_1_	31.79 ± 17.02	29.34 ± 13.20	35.03 ± 20.86	0.5644 _NS_
SD_2_	99.60 ± 34.39	92.89 ± 26.65	108.47 ± 41.39	0.0700 _NS_

NS—not significant.

**Table 6 biology-12-00125-t006:** Analysis of heart rate variability—HRV in the frequency domain for groups divided by the median.

		All	Women	Men	P [w vs. m]
Total Power (ms^2^)	_≤MED._	1867.30 ± 667.92	1911.87 ± 663.92	2000.31 ± 1069.45	0.6228 _NS_
	_>MED._	6232.12 ± 3085.56	4905.17 ± 2027.73	7744.70 ± 3683.16	0.0175 *
LF (ms^2^)	_≤MED._	1029.15 ± 401.93	953.99 ± 368.44	1214.71 ± 570.03	0.2992 _NS_
	_>MED._	3549.42 ± 1905.17	2830.73 ± 1521.06	4389.88 ± 2108.79	0.0008 ***
HF (ms^2^)	_≤MED._	368.42 ± 175.78	415.03 ± 209.75	324.45 ± 148.86	0.2223 _NS_
	_>MED._	2011.14 ± 1408.35	1619.11 ± 910.08	2495.91 ± 1799.09	0.3327 _NS_
SD_1_	_≤MED._	19.81 ± 5.44	19.83 ± 5.54	19.87 ± 5.65	0.9274 _NS_
	_>MED._	44.15 ± 16.03	39.39 ± 11.34	50.19 ± 19.41	0.1774 _NS_
SD_2_	_≤MED._	73.55 ± 16.97	72.19 ± 16.46	76.05 ± 18.99	0.5973 _NS_
	_>MED._	126.46 ± 26.03	114.73 ± 15.26	140.90 ± 30.55	0.0194 *

* *p* < 0.05; *** *p* < 0.001; and NS—not significant.

**Table 7 biology-12-00125-t007:** Correlation coefficients between MIN_div and the selected parameters of HRV in the time domain.

		All		Women		Men	
		MIN_div	MAX_div	MIN_div	MAX_div	MIN_div	MAX_div
SD_NN	_≤MED._	0.1841 _NS_	−0.0116 _NS_	0.0228 _NS_	−0.0894 _NS_	0.3758 _NS_	0.0901 _NS_
	_>MED._	−0.5139 **	−0.1293 _NS_	−0.3519 _NS_	0.0670 _NS_	−0.4466 _NS_	−0.2527 _NS_
rMSSD	_≤MED._	−0.1223 _NS_	−0.1019 _NS_	−0.3473 _NS_	−0.3045 _NS_	0.0945 _NS_	0.19656 _NS_
	_>MED._	−0.5676 ***	−0.0901 _NS_	−0.4468 _NS_	0.2363 _NS_	−0.7758 *	−0.0505 _NS_
pNN50%	_≤MED._	−0.1756 _NS_	−0.0269 _NS_	0.2501 _NS_	−0.2150 _NS_	0.0945 _NS_	−0.1032 _NS_
	_>MED._	−0.4797 **	−0.0720 _NS_	−0.0505 _NS_	0.1558 _NS_	−0.6556 *	−0.2759 _NS_

* *p* < 0.05; ** *p* < 0.01; *** *p* < 0.001; and NS—not significant.

**Table 8 biology-12-00125-t008:** Correlation coefficients between MIN_div; MAX_div and the selected parameters of HRV in the frequency domain.

		All		Women		Men	
		MIN_div	MAX_div	MIN_div	MAX_div	MIN_div	MAX_div
Total Power (ms^2^)	_≤MED._	0.0431 _NS_	−0.0725 _NS_	0.0315 _NS_	−0.0543 _NS_	0.1736 _NS_	0.0285 _NS_
	_>MED._	−0.5174 **	−0.1352 _NS_	−0.2796 _NS_	0.2260 _NS_	−0.4224 _NS_	−0.0945 _NS_
LF (ms^2^)	_≤MED._	0.1560 _NS_	−0.1443 _NS_	0.1789 _NS_	−0.3771 _NS_	0.1428 _NS_	0.0065 _NS_
	_>MED._	−0.4160 *	−0.2364 _NS_	0.3828 _NS_	−0.3353 _NS_	−0.1672 _NS_	−0.0945 _NS_
HF (ms^2^)	_≤MED._	−0.2065 _NS_	0.2465 _NS_	−0.2175 _NS_	0.0728 _NS_	−0.3010 _NS_	0.2483 _NS_
	_>MED._	−0.6440 ***	−0.2100 _NS_	−0.3085 _NS_	0.1124 _NS_	−0.6263 *	−0.2043 _NS_
SD_1_	_≤MED._	−0.1226 _NS_	−0.0986 _NS_	−0.3491 _NS_	−0.2957 _NS_	−0.0945 _NS_	0.1956 _NS_
	_>MED._	−0.5676 ***	−0.0901 _NS_	−0.4468 _NS_	0.2363 _NS_	−0.7758 **	0.0505 _NS_
SD_2_	_≤MED._	0.0060 _NS_	−0.1848 _NS_	−0.1403 _NS_	−0.2052 _NS_	0.2131 _NS_	−0.1428 _NS_
	_>MED._	−0.4853 **	−0.0218 _NS_	−0.2321 _NS_	0.1011 _NS_	−0.3322 _NS_	0.1252 _NS_

* *p* < 0.05; ** *p* < 0.01; *** *p* < 0.001; and NS—not significant.

## Data Availability

Not applicable.

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
