# Peer review of "Heart Rate Variability at Rest Predicts Heart Response to Simulated Diving"

_biology, 2023, doi:10.3390/biology12010125_

Round 1

Reviewer 1 Report

Dear,

Manuscript Number: biology-2111000

Title Manuscript: Heart rate variability at rest predicts heart response to simulated diving

This study investigated the effect of simulated diving on cardiac responses such as heart rate and heart rate variability in 65 healthy men and women with age range of 20-27 yrs. Although this study is an interesting study, but at the moment MAJOR REVISIONS are necessary in order to make it suitable for a final decision for “Biology”.

POINTs of STRENGTH:

1) The effect of simulated diving on cardiac responses such as heart rate and heart rate variability in healthy men and women;

POINTs of WEAKNESS (and/or should be revised to improve the manuscript):

Abstract:

2) The purpose of this study is not specified in the objective section; please specify clearly;

3) The mean age, BMI, gender and as well as number of participants are not specified in the methods section; please clarify;

4) The study protocol for simulated diving is not specified in the methods section; please clarify;

5) Please report the “conclusion section of the Abstract” based on the results obtained from the study.;

6) Please modify the keywords section as follows:

Diving; simulated diving test; Heart rate; Heart rate variability; Healthy individuals.  

Introduction:

7) Please remove the subheadings of “Diving response”, “Heart Rate Variability – HRV” and “Relationship between HRV and cardiac response to diving” in the introduction section;

8) Please modify the abbreviation “AMA” to “Ama diving” in the introduction section. In addition, modify the phrase “heart rate variability analysis (HRV)” to “heart rate variability (HRV) analysis”

9) To use an abbreviation, please write the full name in the first instance and follow it immediately by the abbreviated version in brackets in the manuscript text such as “ANS” in the introduction section and so on;

10) The hypothesis and purpose of this study can be stated in more detail;

2. Materials and methods

11) The type of study is not specified; please clarify;

12) The recruitment process and screening of the study participants, especially inclusion and exclusion criteria should be described in more detail such as number of initial participants, BMI, healthy status-respiratory disorders, blood pressure, free of medications, physical fitness level, VO2max or METs, and so on;

13) Please describe fully the study protocol of simulated diving test.

14) Please explain the assessment of heart rate during simulated diving.

Statistical analyses

15) The significance level of statistical analysis was considered for two-tailed? OR one-tailed? Please specify;

16) Did the authors use statistical software to calculate the sample size? IF YES, please explain and add its name and valid reference in the “statistical analysis” section.

3. Results

17) Please provide the physiological characteristics of the participants in the form of a Table in the results section;

18) Please modify subheading “3.1. Cardiac response to diving” to “3.1. HR response to simulated diving” in the Results section;

19) Based on the HR response to simulated diving, please modify the title of Table 1 to “The HR response to simulated diving in healthy men and women”; Are correct the minimum and maximum diving OR the minimum and maximum heart rate (Min-HR and Max-HR)? Please modify;    

20) The unit of the variables is not specified in the Tables. Please clarify;

4. Discussion

21) As mentioned above, the authors will agree that the limitations section has to be expanded in this study.

5. Conclusions

22) What are the conclusions and implications for the healthy individuals and/ or for clinical studies? And particularly for future studies?;

23) What does this study add to the literature? Please explain and add in the conclusions section.

References

24) References section is not always in accordance with the authors' guidelines. In particular, please check No. 4, 5, and 14 for validation.

Best Regards

12 December 2022

Author Response

Dear Reviewer,

I have decided to attach a revised version of the manuscript taking into account all your comments. Below I wrote answers to your comments, with the exact number of lines where you will find corrections.

Thank you very much for your valuable comments, we are very grateful for pointing out these shortcomings. Please see our responses below.

Kind regards, Krzysztof Malinowski

Abstract:

2) The purpose of the study was specified. Line 32

3) All missing data regarding: age, BMI, gender, number of participants in the methods section has been added. Line 34 ...

4) Study protocol for simulated diving was specified. Line 36 5) Conclusion section of the Abstract was completed. Line 40 6) Keywords were modified. Line 49
Introduction:

7) Subheadings of “Diving response”, “Heart Rate Variability – HRV” and “Relationship between HRV and cardiac response to diving” were removed.

8) Abbreviation “AMA” was modified to “Ama divers. Line 82. Phrase “heart rate variability analysis (HRV)” was modified to “heart rate variability (HRV) analysis” Line 38

9) “To use an abbreviation, please write the full name in the first instance and follow it immediately by the abbreviated version in brackets in the manuscript text such as “ANS” in the introduction section and so on;” - text was checked for this suggestion

10) The hypothesis and purpose of this study was detailed. Line 112-134

2. Materials and methods:

11) Type of study was specified. Line 138

12) Recruitment process and screening of the study participants, especially inclusion and exclusion criteria was described in more. Line 140-146

13) Study protocol of simulated diving test was detailed. Line 155-156

14) The assessment of heart rate during simulated diving test was clarified. Line 171-173

Statistical analyses:

15) The significance level of statistical analysis was calculated for two-tailed Line 200
16) All statistical calculations were performed using statistical package Statistica 10 (StatSoft

Inc., USA, Texas) Line 207

3. Results:

17) Anthropometric data were filled. Line 210
18) “Please modify subheading “3.1. Cardiac response to diving” to “3.1. HR response to simulated

diving” in the Results section” -modified.
19) “Based on the HR response to simulated diving, please modify the title of Table 1 to “The HR

response to simulated diving in healthy men and women” – modified, Line 229

“Are correct the minimum and maximum diving OR the minimum and maximum heart rate (Min-HR and Max-HR)?” – It was decided to use the abbreviations MIN_div and MAX_div to distinguish and avoid confusion with the abbreviations MIN_HR0 and MAX_HR0 referring to minimum and maximum HR at rest.

20) The units of the variables were specified in the Tables.

4. Discussion:

21) Limitations section was expanded. Line 376

5. Conclusions:

22) Conclusions and implications for the healthy individuals and/ or for clinical studies were extended. Line 385-389

23) “What does this study add to the literature?” Line 390-413 References:
24) References section checked

Reviewer 2 Report

In method is not described how long was lasting of diving simulation. Did in your studied cases you have seen evoked any arrhythmias. When calculated Min or Max of HR appear at beginning or end of test. What sense is to talk about HR 9 - 24 per min or other HRV analysis frequency ?

In line 204 are shown significance of results which is not presented in tables. Th same in line 216, 218, 237. 

in many places >MED is presented in small letters, when all statistical indexes are presented in large size.

Author Response

Dear Reviewer,

I have decided to attach a revised version of the manuscript taking into account all your comments. Below I wrote answers to your comments, with the exact number of lines where you will find corrections.

Thank you very much for your valuable comments, we are very grateful for pointing out these shortcomings. Please see our responses below.

Kind regards, Krzysztof Malinowski

1) The time of simulated diving test is in the Line 37 and 156

2) In the current study, arrhythmias occurred rarely and concerned only a few cases (5 times). Line 221

3) “What sense is to talk about HR 9 - 24 per min or other HRV analysis frequency?” – HRV analysis was used as a tool for assessing the influence of the autonomic nervous system on the heart rhythm, and thus determining the role of neurogenic regulation on the cardiac response to diving. the Described frequencies refer to heart rate variability in the frequency domain and refer to short or long term variability that represents the sympathetic and parasympathetic influences on the heart rate.

4) Statistical data relating to comparisons between the groups are marked with symbols referring to statistical significance. Asterisk or NS abbreviation symbols.

5) Abbreviations >MED have been standardized to small letters.

Round 2

Reviewer 1 Report

Dear,

Manuscript Number: biology-2111000

Title Manuscript: Heart rate variability at rest predicts heart response to simulated diving

I am very grateful to the authors for their efforts.

In general, this manuscript has found suitable content after correcting major revisions, and the modified revisions are accepted.

Please modify the spelling mistakes for the final version of the manuscript: 153 (21 yers) and 411 (therelationship).

Best Regards

9 January 2023

Reviewer 2 Report

Thank you for improved version of article.